# Network analysis shows decreased ipsilesional structural connectivity in glioma patients

Lucius S. Fekonja [1,2 ✉], Ziqian Wang [1], Alberto Cacciola [3], Timo Roine[4,5], D. Baran Aydogan[4,6,7], Darius Mewes[1], Sebastian Vellmer[1], Peter Vajkoczy[1] & Thomas Picht[1,2]

Gliomas that infiltrate networks and systems, such as the motor system, often lead to substantial functional impairment in multiple systems. Network-based statistics (NBS) allow to assess local network differences and graph theoretical analyses enable investigation of global and local network properties. Here, we used network measures to characterize glioma-related decreases in structural connectivity by comparing the ipsi- with the contralesional hemispheres of patients and correlated findings with neurological assessment. We found that lesion location resulted in differential impairment of both short and long connectivity patterns. Network analysis showed reduced global and local efficiency in the ipsilesional hemisphere compared to the contralesional hemispheric networks, which reflect the impairment of information transfer across different regions of a network.

[1] Department of Neurosurgery, Charité - Universitätsmedizin Berlin, Berlin, Germany. [2] Cluster of Excellence: "Matters of Activity. Image Space Material", Humboldt University, Berlin, Germany. [3] Department of Biomedical, Dental Sciences and Morphological and Functional Images, University of Messina, Messina, Italy. [4] Department of Neuroscience and Biomedical Engineering, Aalto University School of Science, Espoo, Finland. [5] Turku Brain and Mind Center, University of Turku, Turku, Finland. [6] Department of Psychiatry, Helsinki University and Helsinki University Hospital, Helsinki, Finland. [7] A.I. Virtanen Institute for Molecular Sciences, University of Eastern Finland, Kuopio, Finland. ✉email: lucius.fekonja@charite.de

Gliomas often lead to significant functional impairments that are not always linked to the specific localization of the glioma. While classical theory assumes that local lesions have an exclusively local impact, there is increasing evidence that brain tumors yield not only local structural changes and thus may globally affect the brain[1]. Modern MRI techniques such as diffusion MRI allow us to measure the structural connectivity of anatomically pre-defined brain areas, enabling the representation of brains by brain networks that contain specific brain areas as nodes and the quantified connectivity between nodes as edges. Mapping the effects of brain tumors on spatially distributed brain networks may be essential for a better understanding of anatomical-functional relationships[2]. In recent years, network neuroscience has contributed significantly to mapping brain function to structure and has advanced precision medicine by identifying quantitative biomarkers for assessing brain disease severity[3]. Toolboxes for network-based analyses, such as network-based statistics (NBS), allow connectome-wide non-parametric analyses to identify groups of connections showing a significant effect while controlling the family-wise errors[4]. Although NBS does not provide information on network topology, complex-network measures of global- and local-scale network organization[5,6] have emerged as valuable and reproducible tools for exploiting the topological network architecture of the brain in healthy and diseased subjects[7,8]. Graph theoretical network analysis, applied to structural and functional connectome data in glioma patients, has already shown that network efficiency correlated with cognitive performance in IDH1 wildtype astrocytoma[9] and that alterations in distinct connectome profiles are related to clinical phenotype in newly diagnosed glioma patients[10].

Here, we combine tractography with graph theoretical analysis and NBS to assess tumor-related structural connectome alterations within the ipsilesional hemisphere of glioma patients.

While many studies use functional MRI measurements to determine functional connectivity of the brain, here we use diffusion MRI (dMRI) data and analyze the resulting structural connectivity. In addition, to address the variability in results due to the choice of tractography algorithm, we employed two different algorithms: (i) deterministic and (ii) probabilistic. Moreover, in contrast to many other network analysis studies[11–14], we used a state-of-the-art dMRI processing pipeline that involves constrained spherical deconvolution (CSD) for the estimation of fiber orientation distributions (FODs)[15], anatomically constrained tractography[16] and spherical-deconvolution informed filtering of tractograms[17] within the MRtrix3 open software work frame[18]. The use of different tractography algorithms allowed us to better compare and interpret the results. With this study, we aim to gain detailed insights into how World Health Organization (WHO) grade II–IV gliomas affect cerebral networks and lead to altered connectivity patterns that affect motor and non-motor functions. We hypothesize that asymmetries between ipsi- and contralesional connectivity profiles are related to specific tumor locations that correlate with functional impairment or neurological patient status.

## Results
We investigated the structural network differences in 37 glioma patients, cf. Table 1 and Fig. 1; Please refer to "Methods" for a detailed description). The assessment of MRC, NIHSS and tumor volume, and TMS mapping related RMT determination was feasible in all patients. NIHSS was discarded in three patients due to incomplete data. Enantiomorphic lesion filling, FastSurfer segmentation, iFOD2-based, and SD_STREAM-based connectome construction was possible in each subject.

**Network-based statistics.** To analyze structural differences between the contra- and ipsilesional hemispheres, we apply the threshold-free network-based statistics algorithm (TFNBS, see details to the method in 'Statistical analysis' section below). We obtained 30 iFOD2-related and 19 SD_STREAM-related significant edges by TFNBS on the entire cohort, cf. Fig. 2. Furthermore, we obtained 15 iFOD2-related and 14 SD_STREAM-related significant edges in the precentral subgroup and 15 iFOD2-related and 6 SD_STREAM-related significant edges in the insular subgroup, cf. Fig. 2. Frontal and postcentral groups revealed no significant differences by TFNBS.

We performed Spearman correlation analyses between the strength of the edges revealed as significant by TFNBS and MRC, NIHSS, RMT ratio, tumor volume and WHO grade variables for ipsilesional, contralesional and differences between ipsi- and contralesional matrices (see "Methods", Statistical analysis). The Spearman correlation analyses resulted in 6 (3 iFOD2-based, 3 SD_STREAM-based) false discovery rate (FDR)-corrected significant relationships, cf. Fig. 3.

**Probabilistic tractography.** Based on iFOD2 connectome matrices, there was a significant positive correlation between the streamlines' strength of posterior cingulate—pars opercularis, and NIHSS in the contralesional hemispheres in the precentral group, $rs(13) = 0.79$, $p = 0.0238$. Furthermore, we found a positive correlation of tumor volume and inferior temporal gyrus—fusiform gyrus, $rs(8) = 0.90$, $p = 0.0301$ regarding the differences of ipsi- and contralesional matrices in the insular group. Moreover, we observed a negative correlation of tumor volume and inferior temporal gyrus—fusiform gyrus, $rs(8) = -0.71$, $p = 0.0039$ in the ipsilesional insular group.

## Table 1 Demographic data and clinical characteristics.

| Characteristics | n (%) / M ± SD |
|---|---|
| Gender (n = 37) | |
| Male | 22 (59) |
| Female | 15 (41) |
| Age | 48.24 ± 16.47 |
| Glioma WHO grading | |
| II | 13 (35) |
| III | 10 (27) |
| IV | 14 (38) |
| Glioma location (n = 37) | |
| Left hemisphere | 16 (43) |
| Right hemisphere | 21 (57) |
| Subgroup | |
| Precentral subgroup | 16 |
| Postcentral subgroup | 15 |
| Insular subgroup | 8 |
| Frontal subgroup | 3 |
| Glioma volume (cm³) | 24.97 ± 23.84 |
| MRC scale (n = 37) | |
| 2 | 1 (5) |
| 3 | 1 (5) |
| 4 | 12 (35) |
| 5 | 23 (55) |
| RMT (V/m, n = 37) | |
| Pathological hemisphere | 35.13 ± 7.61 |
| Healthy hemisphere | 34.59 ± 6.81 |
| Pathological/healthy hemisphere | 1.02 ± 0.166 |
| NIHSS (n = 34) | 0.94 ± 1.50 |

*MRC scale* Medical Research Council scale, from 0 to 5, *RMT* resting motor threshold, *NIHSS* National Institutes of Health Stroke Scale.

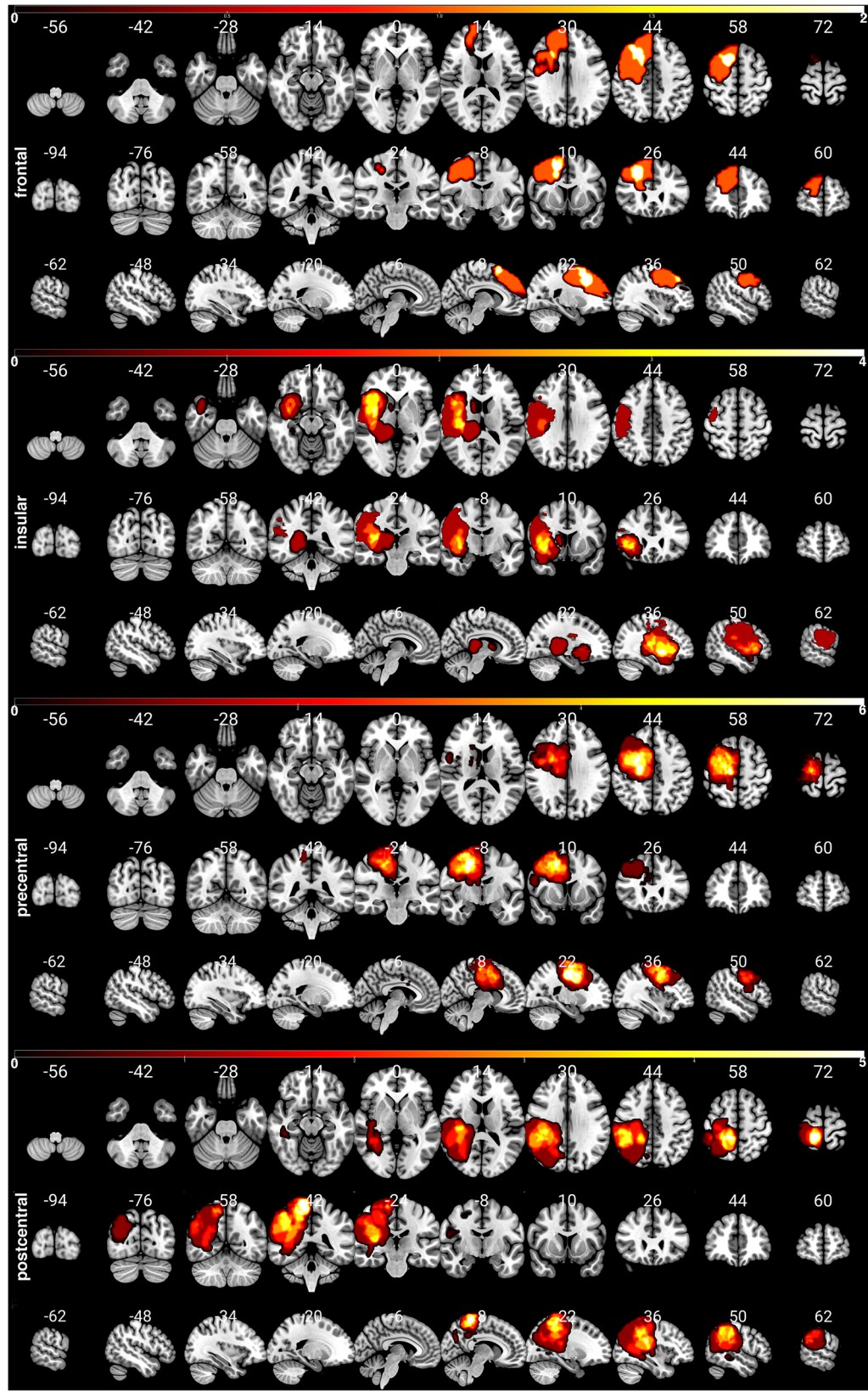

**Fig. 1 Distribution of the subgroups' patients' lesions (frontal, insular, precentral, and postcentral).** The color bar indicates the occurrence of lesions per voxel. To enable a clear comparison of lesion location, lesions of the left hemisphere were mapped to the right hemisphere.

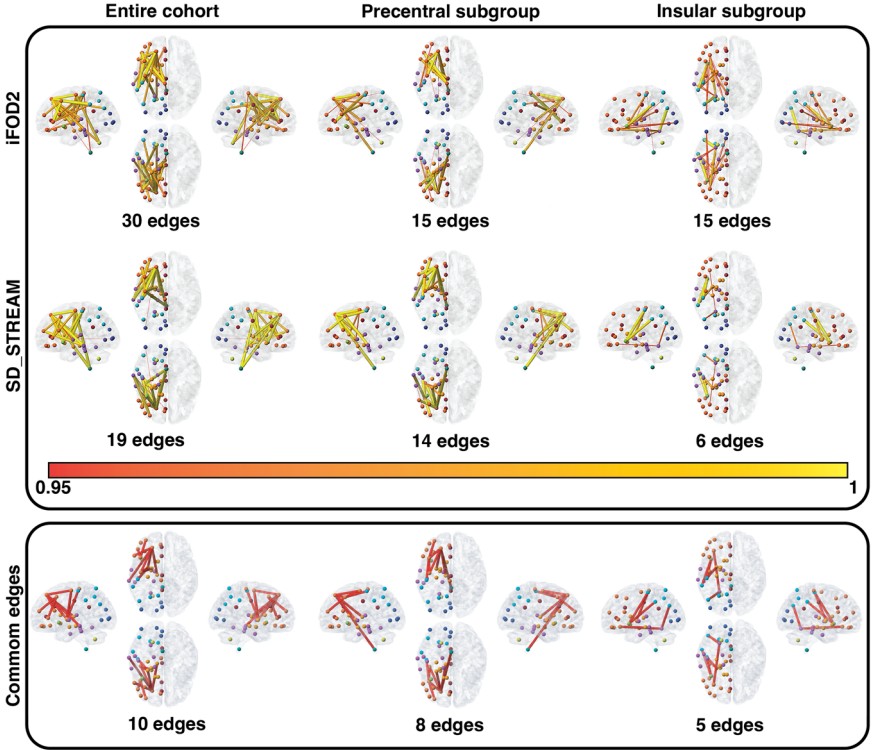

**Fig. 2 TFNBS results visualizing the significant differences between ipsi- and contralesional connectome matrices.** The heatmap indicates the FWE-corrected significance (1-p, only significant edges are displayed with *p*-values < 0.05). The node colors reflect the atlas parcellation (Desikan–Killiany–Tourville) color scheme. Common edges: Significant edges that were significant in both tractography (iFOD2 & SD_STREAM) algorithms (*p*-values < 0.05), cf. Supplementary Information File.

**Deterministic tractography**. Based on SD_STREAM connectome matrices, there was a significant positive correlation between the weight of inferior parietal gyrus—fusiform gyrus and WHO grade regarding the difference between contra—and ipsilesional hemispheres in the insular group, rs(8) = 0.88, $p = 0.0226$. In addition, we observed a significant positive correlation between the streamline strength of inferior parietal gyrus and fusiform gyrus in relation to WHO grade in the contralesional hemispheres in the insular group, rs(8) = 0.88, $p = 0.0226$. Furthermore, we observed a significant negative correlation between the streamline strength of putamen-postcentral gyrus and WHO grade in the ipsilesional hemispheres and the entire cohort, rs(37) = −0.48, $p = 0.0461$.

**Complex network analysis**. Graph theoretical analysis showed significant correlations of the RMT ratio with small worldness and local efficiency in contralesional hemispheres (both rs(37) = −0.373, $p = 0.0228$), but not with any clinical measure (i.e., MRC, NIHSS, WHO grade, tumor volume) for deterministic and probabilistic tractography results. There were no significant differences for assortativity, nodal degree, hierarchy, nodal efficiency, rich club, and small worldness between contra- and ipsilesional hemispheres. However, for some global network metrics, there were significant differences between contra- and ipsilesional hemispheres, cf. below, *Complex network analysis in relation to probabilistic tractography* and *complex network analysis in relation to deterministic tractography*). iFOD2-based differences resulted in a correlation of MRC with hierarchy (rs(37) = 0.372, $p = 0.02339$) and local efficiency (rs(37) = 0.349, $p = 0.03432$). SD_STREAM-based differences resulted in a correlation of MRC and local efficiency (rs(37) = 0.326, $p = 0.0489$) and NIHSS and local efficiency (rs(37) = −0.361, $p = 0.03606$), cf. Fig. 4.

**Complex network analysis in relation to probabilistic tractography**. Based on iFOD2 connectome matrices, there was a significant decrease in the global efficiency of ipsilesional hemispheres (M = 4955, SD = 429) compared to the contralesional hemispheres (M = 5239, SD = 436), t(36) = 3.3, $p = 0.0024$ as well as a significant decrease in the local efficiency of pathological hemispheres (M = 5006, SD = 411) compared to the healthy hemispheres (M = 5241, SD = 446), t(36) = 3.05, $p = 0.0043$, cf. Fig. 4.

**Complex network analysis in relation to deterministic tractography**. Regarding SD_STREAM-based connectome matrices, there was a significant decrease in the global efficiency of ipsilesional hemispheres (M = 4584, SD = 507) compared to the contralesional hemispheres (M = 4950, SD = 415), t(36) = 3.7, $p = 0.00078$ and a significant decrease in the local efficiency of ipsilesional hemispheres (M = 4948, SD = 434) compared to the healthy hemispheres (M = 5176, SD = 482), t(36) = 3.34, $p = 0.0019$, cf. Fig. 4.

Furthermore, SD_STREAM and iFOD2 matrices were found to be strongly positively correlated in local and global efficiency in ipsi- and contralesional groups. Global efficiency contralesional: rs(72) = 0.92, $p < 0.000$, Global ipsilesional: rs(72) = (0.86, $p < 0.000$, local efficiency contralesional: rs(72) = 0.93, $p < 0.000$ and local efficiency ipsilesional: rs(72) = 0.88, $p = 0.000$.

In addition to the correlations, local efficiency showed no differences between the two tractography algorithms with respect to the same hemisphere, but significant differences between ipsi- and contralesional hemispheres appeared, whereas global efficiency showed differences between the two hemispheres as well as between the two tractography algorithms with respect to the same hemisphere, cf. Fig. 4.

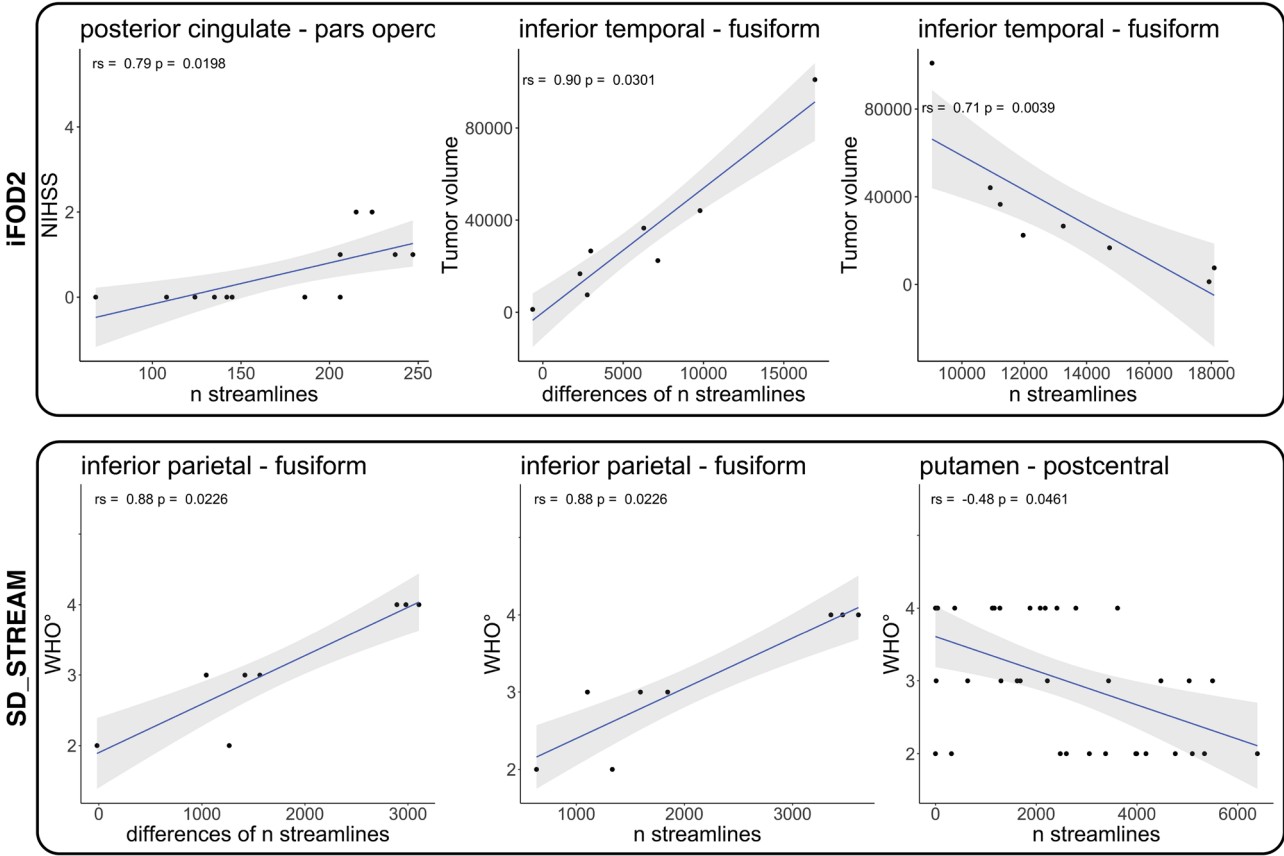

**Fig. 3 Spearman correlations of TFNBS-selected significant edges.** Line plots of significant FDR-corrected Spearman correlations between TFNBS-selected significant edges and NIHSS, WHO grade or tumor volume for both tractography algorithms (iFOD2 & SD_STREAM).

## Discussion

The last decades have been marked by significant advances in the characterization of brain morphology, function, and brain disorders using connectomics[19]. The localizationist theory has been assimilated into associationist models that see the brain organized in parallel distributed networks[1]. Primary sensory and motor functions are considered more focally localized[20,21], while higher cognitive functions are discussed as organized in large-scale[22] distributed networks. Brain functions and complex behavior are thought to arise from parallel processing and integration performed by large-scale distributed networks rather than single epicenters[23-25]. In this scenario, connectomics and graph theory could prove as powerful tools to map and relate the architecture of brain networks from structure to function and to identify specific neural substrates associated with dysfunction[6,26].

We constructed structural connectomes from glioma patients to investigate the effects of tumors on ipsi- and contralesional networks. To provide a comprehensive overview and better characterize our results, we used two different tractography algorithms, namely the probabilistic CSD-based iFOD2 and the deterministic CSD-based SD_STREAM, which resulted in two distinct types of structural connectomes.

Although several studies have shown that tumor-induced changes in the WM could be analyzed using dMRI-based methods[11,27], to our knowledge, this is the first study characterizing quantitatively and qualitatively structural connectivity changes in tumor patients at the hemispheric network level.

An interesting finding of this study is the impairment of the ipsilesional structural connectivity compared to the contralesional hemisphere. Earlier studies have shown an increase in ipsilesional raw fiber count[28], as well as an increased functional connectivity

of the hippocampus and antero-medial portion of the posterior cingulate in glioma patients compared to healthy controls[29]. However, from a structural connectivity perspective, these findings are likely to be more widely present than the morphological effects of glioma, including tract displacement, edema, blood-brain-barrier disruption, necrosis, or degradation of the surrounding cerebral tissue[30,31]. In line with this hypothesis, TFNBS analysis identified significant disconnected subnetworks involving fronto-frontal, fronto-parietal, and fronto-insular edges in the ipsilesional hemispheres compared to the contralateral ones. Furthermore, we observed a significant negative correlation between streamline strength of the putamen-postcentral gyrus connectivity in the ipsilesional hemisphere and WHO grade in the entire cohort. This result suggests that higher WHO grade gliomas are more likely to impair the cortico–subcortical connections regardless of their location, thus further reinforcing the idea that patterns of altered structural connectivity depend on the WHO grade[10,32].

As earlier studies have demonstrated local effects of gliomas on their surrounding microstructural integrity of the WM[33,34], we aimed at investigating whether glioma locations may affect unique subsets of nodes and edges in the ipsilesional network. While gliomas affect brain regions and connections near the lesion, brain connectivity changes may also occur distally from the lesion, either because of the resulting local dysconnectivity or as a neuroplasticity mechanism taking place during tumor growth. In line with this hypothesis, our subgroup analysis showed that lesion location (for instance, pre-central and insular) affected connectivity patterns that belong to distinct subnetworks. We obtained several iFOD2- and SD_STREAM-related significantly affected edges in the precentral and in the insular

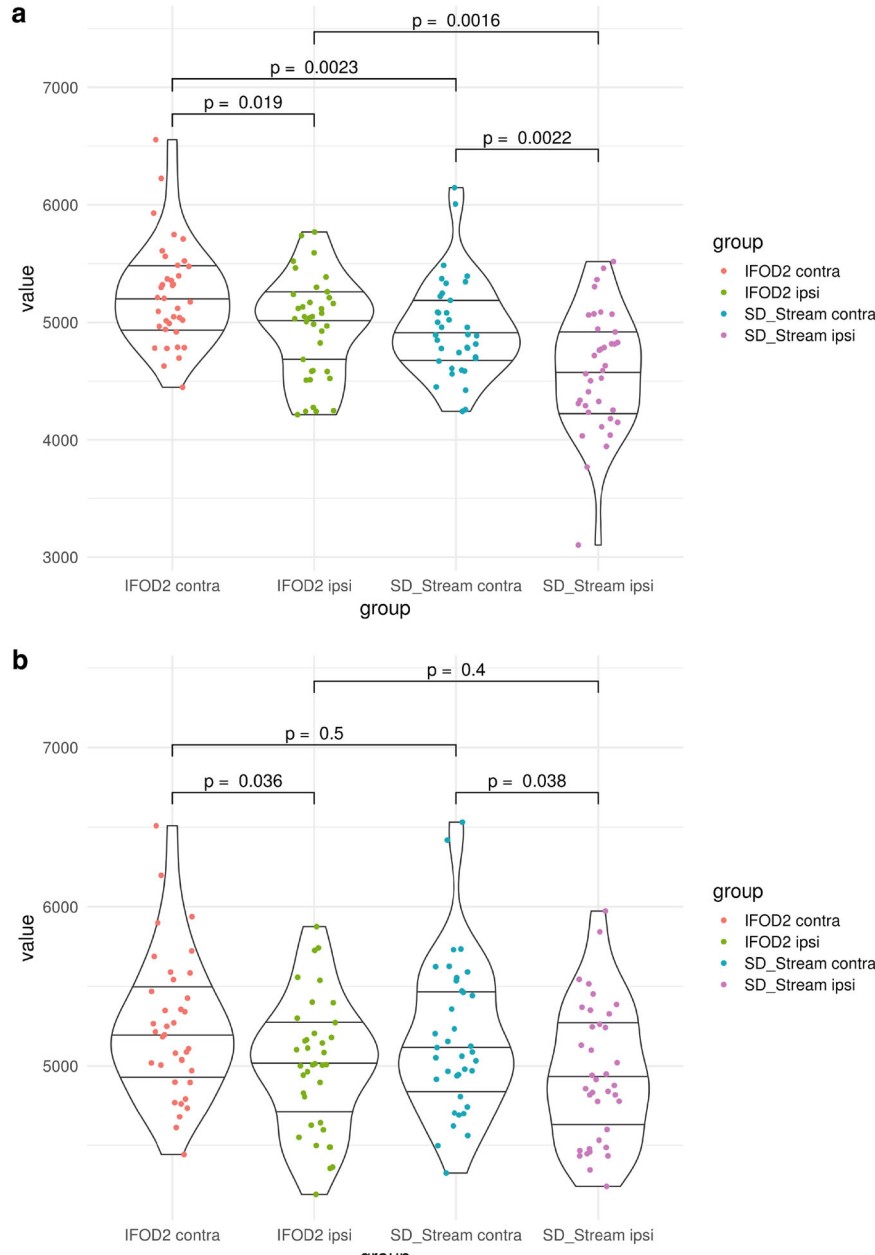

**Fig. 4 Wilcoxon tests for efficiency measures.** Violin plots of Wilcoxon tests for global (**a**) and local efficiency (**b**) by testing the differences of ipsilesional (*_ipsi) and contralesional (*_contra) hemispheres in relation to the tractography algorithms.

subgroup. By contrast, no significant TFNBS differences emerged for the frontal and postcentral groups, neither using iFOD2- nor SD_STREAM-based connectome matrices. Additionally, we found a significant positive correlation between streamline strength of the posterior cingulate—pars opercularis edge of the contralesional hemisphere and NIHSS in the precentral subgroup, as well as a significant positive correlation between streamline strength of inferior parietal gyrus—fusiform gyrus and WHO grade in the contralesional hemisphere in the insular subgroup. These findings may suggest that the presence of increased structural connectivity in the contralesional hemisphere is correlated both with worse clinical conditions and higher-grade gliomas, suggesting possible compensatory adaptive changes of connectomic profiles in contralesional networks in patients with unilateral gliomas[10,35]. In addition, we found that tumor volume was correlated with the inferior temporal gyrus-fusiform gyrus

edge. A larger tumor volume in the insular region likely impairs the connection strength in the ipsilesional hemisphere in this patient subgroup. Furthermore, the fusiform gyrus, particularly its anterior portion, has been shown to be strongly connected to the inferior temporal gyrus, which, as part of the human ventral visual cortex, plays a role in higher-order visual processing such as face perception and object recognition[36,37]. Overall, our results not only demonstrate impaired ipsilesional structural connectivity in glioma patients, but also suggest that a possible neural basis of higher-order symptoms may occur due to the distributed organization of function.

TFNBS offers high sensitivity in detecting altered patterns of connectivity in a network. However, TFNBS data are not specific to any network topology measure, i.e., they cannot offer information related with a particular property of the topology that differs between the hemispheres, despite the identified sub-

networks displaying significant between-hemispheres differences. Therefore, the deeper understanding of brain topology and the extent to which a network holds certain topological characteristics, e.g., integration and differentiation, are important to find key elements supporting the structural connectivity involvement in glioma patients. About that, complex network topology analysis of the structural connectomes allows for investigating both the global and local topological organizations, as well as specific connections between the regions. Hence, to provide a comprehensive characterization of the hemispheric structural network changes, we computed especially two common network measures, namely global and local efficiency. Both iFOD2-based and SD_STREAM-based structural connectomes showed reduced global and local efficiency in the ipsilesional hemisphere compared to the contralesional hemispheric networks. Global efficiency is a measure that is inversely related to the topological distance between nodes and quantifies how efficiently the information is exchanged within the network. The reduced values of global efficiency suggest less efficient pathways from one brain area to another and, consequently, lower levels of integration in the ipsilesional hemispheric network.

The average local efficiency quantifies the ability of fault tolerance of the network measuring the information exchange of the subnetwork consisting of itself and its direct neighbors. Therefore, the lower values of local efficiency found in the ipsilesional hemisphere network suggest that the structural brain network of glioma patients is topologically organized to minimize segregation of neural processing.

Reduced global and local efficiency reflects the impairment of information transfer across different regions of derived networks, which is likely linked to the pathological involvement of both long- and short-range connections. In contrast with these findings, a recent publication showed increased global and local efficiency in the ipsilesional hemispheric network, identifying unique sets of nodes with changes in network efficiency depending on lesion location[34].

As tractography has been discussed to provide high rates of false-positive[38,39] and false negative[40] streamlines and no gold-standard approaches have been identified in general, we used two different tractography algorithms to confirm our findings— probabilistic CSD-based iFOD2 and deterministic CSD-based SD_STREAM. Both algorithms identified the involvement of similar, ipsilesional decreased structural connectivity patterns, with the only difference in the resulting number of decreased edges, confirming the plausible anatomical reliability of our findings, cf. Fig. 2. Here, we found SD_STREAM and iFOD2 to be strongly positively correlated in local and global efficiency both in ipsi- and contralesional hemispheres, however, the absolute values of these measures were found to be significantly different between the two approaches. Indeed, despite probabilistic tractography approaches have been shown to outperform deterministic tracking in reaching the full extent of the bundles[41], the latter are still widely used in clinical settings, and moreover, they are used based on diffusion tensor imaging (DTI). Deterministic tractography algorithms proceed by stepping along the principal direction of diffusion, and thus they do not address uncertainty in fiber orientation. CSD-based probabilistic algorithms on the other hand, assume a distribution of possible orientations for propagation taking into account uncertainty in streamline segment orientation. While this may result in a larger number of false-positive streamlines (with relatively low streamline density), probabilistic algorithms can identify tract segments that are not reconstructed by the deterministic approach, thus potentially reducing the risk of underestimating the real extent of the tracts, i.e., reducing false-negatives[41]. Finally, with regards to connectomics and graph theory, our findings of a linear positive correlation between network measures derived by deterministic and probabilistic CSD-based tractography are in line with previous studies showing significant correlations between link-wise intraclass correlation coefficients from both methodologies[42]. Altogether, while our deterministic and probabilistic tractography-based TFNBS and network measures findings have noticeable differences, they also suggest a certain level of consistency in anatomical reproducibility. Importantly, we did not find significant differences in local efficiency with respect to the two tractography algorithms, cf. Fig. 4, which further strengthens our finding of decreased local efficiency in the ipsilisional hemisphere.

**Clinical correlations.** Available data in the literature left the problem, whether a correlation between clinical scores, histopathology, RMT and network topology measures exists, partially unsolved. Even though global and local efficiency provided a between-hemispheric differentiation, NIHSS, WHO grade, RMT values were poorly correlated. This may be related to the fact that macro- and microscopic brain damage, as in glioma patients, possibly leads to a comparable impairment of network measures in terms of connectivity strength reduction and reorganization phenomena throughout the ipsilesional hemisphere.

In terms of limitations, FastSurfer is not trained with images generated with enantiomorphic filling and thus might have performed differently to using healthy subject data as input. However, FastSurfer employs reliable landmarks while parcellating brain images and has been shown to outperform FreeSurfer with respect to runtime, test-retest reliability, and sensitivity. Tractography methods suffer from a range of limitations that make its routine use problematic. It is well known that tractograms contain false positive[43] and false negative[40] streamlines[44]. In addition, tractography cannot distinguish between afferent and efferent connections, and streamlines may terminate improperly[45]. Furthermore, the use of SIFT in pathological connectomes is currently being discussed[46–49], however, we did not observe any disadvantages. The dMRI data used for this study consists of a typical clinical single-shell acquisition, and is thus suboptimal due to incomplete attenuation of apparent extra-axonal signal[50]. Note that all patients received preoperative steroids to reduce edema, which could possibly cause a confounding effect. However, there is evidence that edema have no strong influence on tractography results as shown in the referenced articles[51].

Moreover, structural brain asymmetries appear to be present even in healthy subjects. A study regarding structural network topology showed that the right hemisphere network is less efficient than that of the left hemisphere[52]. Specifically, the right hemisphere showed higher values of betweenness centrality and small-worldness, indicating a less optimal organization for information processing and a more random configuration compared with left-hemispheric networks. In contrast, a further study found higher global efficiency in the right hemisphere compared with the left hemispheric network[53]. Finally, the small sample sizes for insular ($n = 8$) and frontal ($n = 3$) lobe subgroups make these data susceptible to both outliers and false negatives.

We would like to mention that in our study we used the older 2007[54] WHO classification instead of the one from 2016 or 2021[55,56], which considers grade 2 and 3 gliomas as part of the same spectrum for which molecular markers (e.g., IDH, 1p/19q, TERT promoter and EGFR amplification) are of particular relevance. Thus, the correlations performed here could result in erroneous findings and should be interpreted with caution because, according to the current WHO CNS5 classification[56], neoplasms are graded within types instead across different tumor types, meaning that for instance an IDH-mutant astrocytoma can

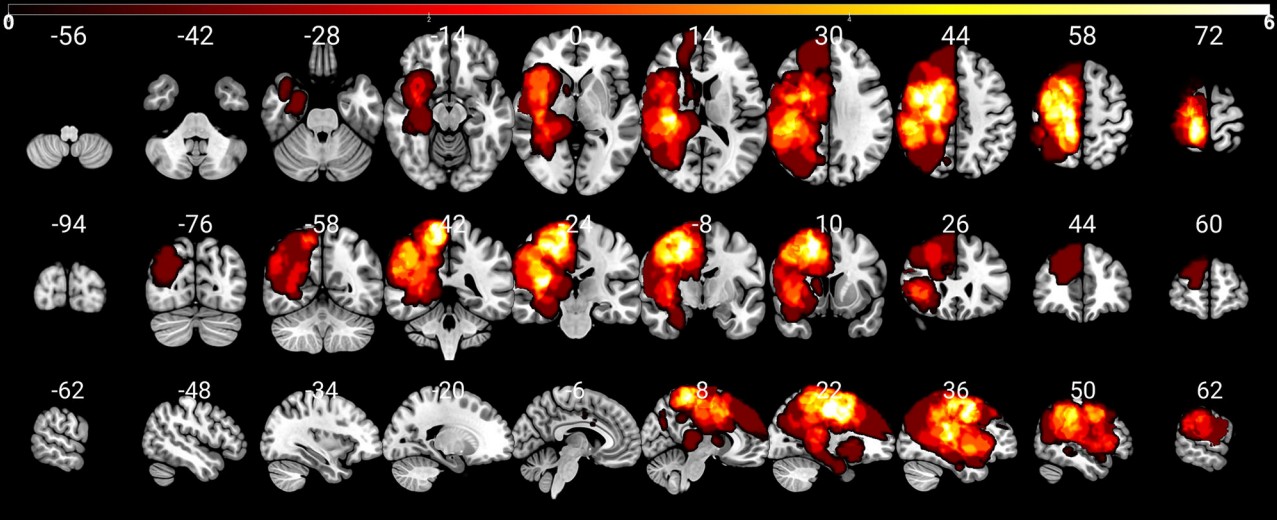

**Fig. 5 Distribution of the patients' lesions.** The color bar indicates the occurrence of lesions per voxel (white = high quantity, 0–6). To enable a clear comparison of lesion location, lesions of the left hemisphere were mapped to the right hemisphere. The numbers on top of the slices show their position in MNI space.

be grade 2, 3 or 4. The grading of CNS tumors has long differed from other non-cerebral neoplasms, as different gradings are used for brain and spinal cord tumors[57]. WHO CNS5 has approximated the grading of non-cerebral neoplasms in the grading of CNS tumors, but has retained aspects of the traditional grading of CNS tumors, as this grading is firmly established in neuro-oncology practice. Two specific aspects of CNS tumor grading have been changed for the WHO CNS5: Arabic numerals are used (instead of Roman numerals) and neoplasms are classified within types instead of between different tumor types[58]. However, inaccurate malignancy classification would occur in very few patients in this study, and therefore this effect may be limited in the correlation analysis.

## Conclusions

In the present study, we showed altered ipsilesional connectivity in patients with unilateral gliomas. TFNBS analysis identified significant disconnected subnetworks involving fronto-frontal and fronto-insular connections in the ipsilesional hemisphere compared to the contralateral one. Our subgroup analysis also showed that the lesion location (e.g., pre-central and insular) affected connectivity patterns that belong to distinct and peculiar subnetworks, thus highlighting the pivotal role of lesion location in driving corresponding connectivity changes. Such connectivity changes were accompanied by reduced global and local efficiency of the ipsilesional network, suggesting tumor-related altered information transfer, which is due to the pathological involvement of both long- and short-range connectivity patterns and disturbed network integration. Moreover, we observed a correlation between the difference of the matrices in terms of hierarchy as well as local efficiency and functional impairment scales, such as MRC and NIHSS. Additionally, deterministic and probabilistic connectome matrices correlated strongly positively in local and global efficiency in ipsi- and contralesional groups. The integration of connectomics into clinical applications is of paramount importance and provides a novel perspective in the neurooncological scenario having the potential to revolutionize personalized medicine and therapy. Indeed, studying structural connectivity in glioma patients through the lens of network neuroscience may guide and improve tumor resection while preserving important nodes and edges which are located more distant from the lesion

but also involved in motor and cognitive functions. Finally, network neuroscience represents an important computational approach to better understand glioma-induced structural changes and contributes to our understanding on the relationship of network topology to motor function.

## Methods

**Patient selection**. We included $n = 37$ left- and right-handed adult presurgical patients in this study (15 females, 22 males, the average age was 48.24, SD = 16.47, age range 20–78). Only patients with an initial diagnosis of unilateral WHO grade II, III & IV gliomas (13 WHO grade II, 10 WHO grade III, 14 WHO grade IV) and without a midline shift in structural images were included, cf. Table 1 and Fig. 5. All gliomas were infiltrating M1 and/or showing adjacency to the corticospinal tract (CST), either in the left ($n = 16$) or right ($n = 21$) hemisphere, Fig. 1. Patients with recurrent gliomas, previous radiochemotherapy, non-glial tumors, or frequent generalized seizures (more than one per week) were not considered.

**Clinical assessment**. We used the National Institutes of Health Stroke Scale (NIHSS) to objectively quantify the impairment caused by the glioma[59]. The NIHSS includes the areas of level of consciousness, eye movements, integrity of visual fields, facial movements, arm and leg muscle strength, sensation, coordination, language, speech, and neglect. Each impairment is scored on an ordinal scale of 0–2, 0–3, or 0–4, with the scores adding up to a total score of 0–42[59]. Besides NIHSS, we used the British Medical Research Council grade (MRC) to assess motor status, where 0 means no muscle activation and 5 means normal muscle strength[60].

**Navigated TMS**. All patients were examined with navigated transcranial magnetic stimulation (nTMS) to perform preoperative functional mapping[61,62]. nTMS is performed by placing a handheld electromagnetic coil on the subjects' skull to excite neurons and provoke motor evoked potentials (MEP) which are recorded using a connected electromyography (EMG) unit[63,64]. Depending on the pathology´s location the EMG activity of different muscles is measured: commonly utilized muscles are the abductor pollicis brevis, first dorsal interosseus (FDI), and abductor digiti minimi for the upper extremity. When examining the motor function of the lower extremity, commonly used muscles are the tibialis anterior and the abductor hallucis. First, the *hot spot*[65] of the FDI muscle was identified by applying TMS in a dense grid and with different coil rotations to achieve the best topographic accuracy[62]. Then, the resting motor threshold (RMT, in V/m), defined as the lowest stimulation intensity sufficient to induce a MEP in at least 5 out of 10 stimulations (≥50 μV), was determined at the top of the cortex for each hemisphere. Peritumoral mapping was then performed for the upper (stimulation intensity: 110% RMT) and lower (median stimulation intensity: 130% RMT) extremities[66]. Finally, mapping (stimulation intensity: 105% RMT) was performed to specifically outline the primary motor cortex along the precentral gyrus.

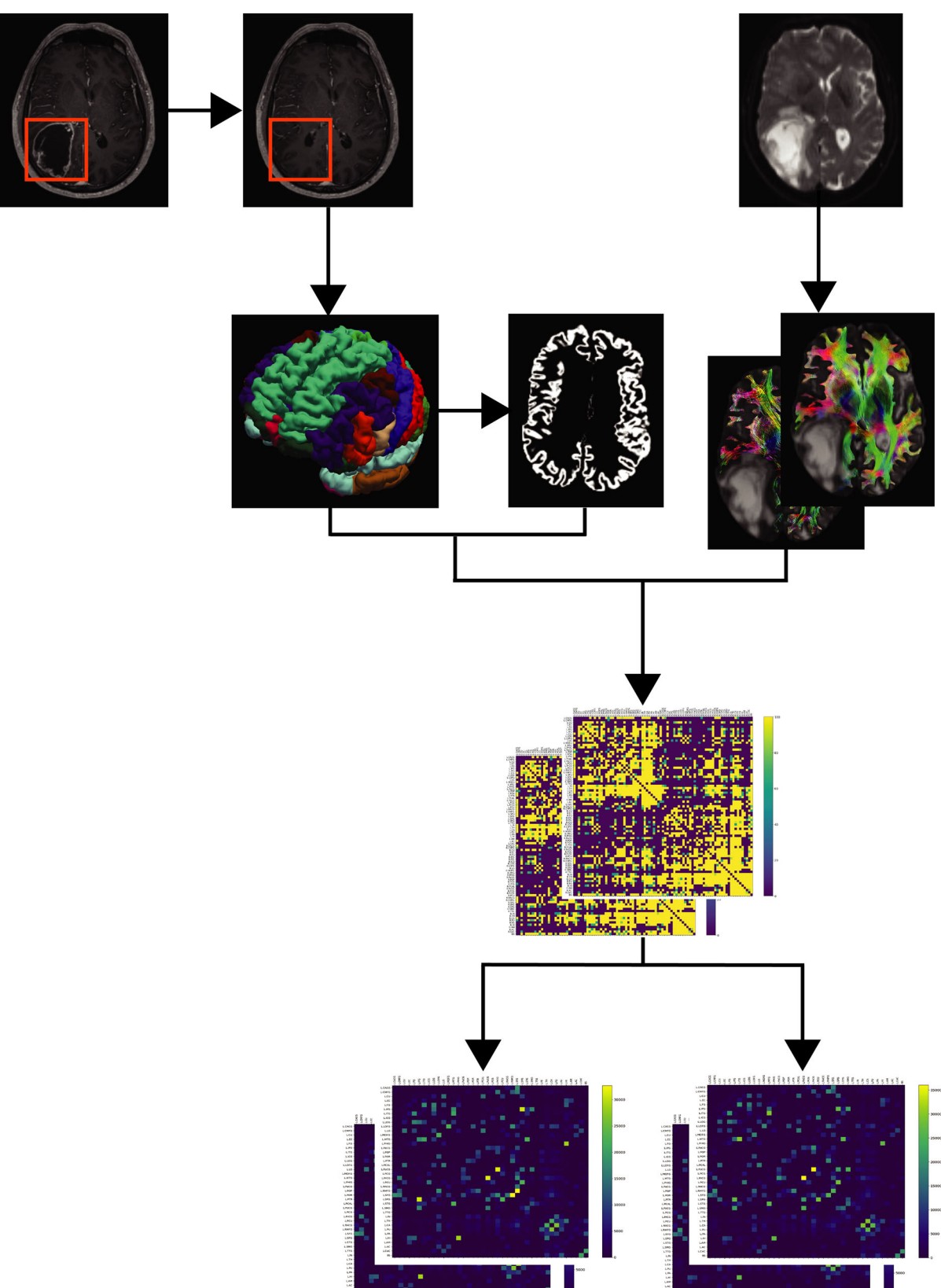

**Fig. 6 Network reconstruction workflow.** Tumors were masked and used for enantiomorphic filling of the T1-weighted structural images, which were then processed by the FastSurfer pipeline to obtain subject-specific parcellations and surface reconstructions. After whole brain tractography within the ACT framework of 50,000,000 streamlines per subject, tractograms were filtered to 10,000,000 streamlines per subject. Further, symmetric connectome matrices were generated and separated into contra- and ipsilesional matrices.

**MRI data acquisition**. Preoperative MRI data were acquired on a Siemens Skyra 3T scanner (Erlangen, Germany) equipped with a 32-channel receiver head coil at Charité University Hospital, Berlin, Department of Neuroradiology. These data consisted of a high-resolution contrast enhanced T1-weighted structural scan (TR/TE/TI 2300/2.32/900 ms, 9° flip angle, 256 × 256 matrix, 1 mm isotropic voxels, 192 slices, acquisition time: 5 min) and a single shell dMRI $2 \times 2 \times 2$ mm$^3$ voxels, 128 × 128 matrix, 60 slices, 3 b0 volumes) image data set, acquired at $b = 0$ and 1000 s/mm$^2$ with 5 and 30 volumes respectively, for a total acquisition time of 12 min. Additionally, T2-weighted and 3D fluid-attenuated inversion recovery (FLAIR) and subtraction sequences were performed.

**T1-weighted structural MRI preprocessing**. All T1-weighted images were registered to the dMRI data sets using the Linear Image Registration Tool (FLIRT) of FMRIB Software Library (FSL, v6.0)[67]. Before obtaining a whole-brain parcellation scheme, the gliomas were manually segmented using Insight Toolkit (ITK) snap, with additional reference to T2 and FLAIR images[68]. After tumor segmentation, the Clinical Toolbox for Statistical Parameter Mapping (SPM) was used for enantiomorphic lesion filling[69,70]. All T1-weighted images were registered to the dMRI data sets using the Linear Image Registration Tool (FLIRT) of FMRIB Software Library (FSL, v6.0)[67]. Before obtaining a whole-brain parcellation scheme, the gliomas were manually segmented using Insight Toolkit (ITK) snap, with additional reference to T2 and FLAIR images[68]. After tumor segmentation, the Clinical Toolbox for SPM was used for enantiomorphic lesion filling[69,70]. After that, the structural T1-weighted images were processed using FastSurfer's[71] deep learning-based processing of structural human brain MRI data, replicating FreeSurfer's anatomical segmentation[72,73]. FastSurfer outputs subject-specific anatomical segmentations including surface reconstructions and cortical parcellations, following the Desikan–Killiany–Tourville (DKT) atlas, resulting in 76 gray matter nodes[74,75]. All results were visually inspected before subsequent computations. Prior to tractography, the structural T1-weighted images were used to generate a five-tissue-type (5TT) image based on Hybrid Surface and Volume Segmentation (HSVS), by using FastSurfer output and FSL tools[16,18,71,72,76]. Following this step, lesion masks of voxels were manually set to the pathological tissue volume fractions in the 5TT images, cf. Fig. 6.

**Diffusion MRI preprocessing and tractography**. The preprocessing of dMRI data included the following and was performed within MRtrix3[18,11] in order: denoising[77,78]: denoising[78], removal of Gibbs ringing artefacts[79], correction of subject motion[80], eddy-currents[81] and susceptibility-induced distortions[82] in FSL[76], and subsequent bias field correction with ANTs N4[79,83], correction of subject motion[80], eddy-currents[81] and susceptibility-induced distortions[82] in FSL[76], and subsequent bias field correction with ANTs N4[83]. Each dMRI data set and processing step was visually inspected for outliers and artifacts. We defined a threshold of scans with more than 10% outlier slices due to excessive motion, however, this was not exceeded in any patient. We upsampled the dMRI data to a 1.3 mm isotropic voxel size before computing FODs to increase anatomical contrast and improve downstream tractography results and statistics[84]. For voxel-wise modeling we used a robust and fully automated and unsupervised method. This method allowed to obtain tissue-specific response functions for white and gray matter and cerebrospinal fluid (CSF) from our data using spherical deconvolution for subsequent use in multi-tissue CSD-based tractography[85–87].

**Streamline tractography**. For tractography, two tracking algorithms were used. Probabilistic tractography was performed with the 2nd-order integration over fiber orientation distributions (iFOD2) algorithm and additional usage of the anatomically constrained tractography (ACT) framework using the 5TT image[15,16,18]. Tracking parameters were set to an FOD amplitude cutoff value of 0.06, a streamline minimum length of 5 × voxel size, and a maximum streamline length of 250 mm. For each tractogram, we computed 50,000,000 streamlines. Further streamline tractography parameters included backtracking, to allow tracks to be truncated and re-tracked if a poor structural termination was encountered, cropping streamline endpoints as they cross the gray matter–white matter interface, and determining seed points dynamically[88]. Subsequently, the whole-brain tractograms were filtered to 10,000,000 streamlines with spherical-deconvolution informed filtering of tractograms (SIFT) such that the streamline densities matched the FOD lobe integrals[17,46]. The lesion masks were used to exclude possible underlying streamlines, cf. Fig. 6. Each tractogram was visually inspected for proper streamlines generation after initial tractogram reconstruction, after SIFT, and after exclusion of streamlines underlying the lesion masks.

In addition to probabilistic tractography, we performed deterministic tractography with the SD_STREAM algorithm, which is as well based on FOD input. The ACT framework was employed here as well. 4th-order Runge–Kutta integration was used to eliminate curvature overshoot[17,89]. Tracking parameters included a 45° angle, a .625 step size, a maximum streamline length of 250 mm and an FOD cutoff value of .06. As above, for each tractogram, we computed 50,000,000 streamlines, which were filtered to 10,000,000 streamlines by SIFT[17].

**Connectome construction**. After whole brain tractogram generation and filtering, we obtained connectome matrices by mapping the streamlines based on their assignments to the node-wise endpoints defined in the FastSurfer based DKT parcellation[71,74,75]. Furthermore, we modified the lookup table (LUT) by adding brainstem and left and right cerebellum labels from Freesurfer's LUT to FastSurfer's DKT LUT to obtain nodes (left and right cerebellum and brainstem) in the connection matrix that represent the cerebral base of the primary motor area connectivity. This resulted in a subject-specific weighted, undirected network represented as a symmetric 79 × 79 adjacency matrix. In this matrix, each node is represented by a DKT area and each edge as the node-wise structural connectivity. The metric of connectivity quantified in the connectome matrix is the number of streamlines[18]. For network analysis, in order to identify structural connectivity changes between the ipsi- and contralesional hemispheres, the whole-brain connectome matrices were split with a MATLAB (R2018b) script into ipsilesional and contralesional matrices representing the two hemispheres separately. This split resulted in a temporary 39 × 39 adjacency matrix and after inclusion of the brainstem in a 40 × 40 adjacency matrix for each hemisphere. Interhemispheric edges were excluded to rule out the confounding effect of the opposing hemisphere in subsequent analyses.

**Statistics and reproducibility**. Statistical analysis was performed by MRtrix3[18] for connectome group-wise statistics at the edge level using non-parametric permutation testing via the threshold-free network-based statistics (TFNBS) algorithm[90]. This algorithm replaces each connectome element $\mathbf{M}_{i,j}$ by its corresponding TFNBS score that is calculated as follows. For each connectome node $ii$, we count the number of connections $e(h)$ to other nodes $j$ that exceed a variable threshold $h$. The TFNBS score is the integral of the product $e(h)^{0.4}h^3 dh$ over thresholds from zero to the value of the considered element $\mathbf{M}_{i,j}$. We performed a permutation test with a default $n = 5000$ shuffling of data for nonparametric statistical inference. The hemispheres of human brains are naturally characterized by structural differences even if no lesion is present. To account for such hemispheric differences, we added the hemispheric tumor position as a covariate[52]. TFNBS provides multiple hypothesis testing at the level of interconnected subnetworks and controls family-level errors (FWE) in the performance of analyses associated with a particular effect or contrast of interest[90]. TFNBS overcomes some of the limitations of the generic procedure FDR, which computes statistical tests and the corresponding $p$-value independently for each link and considers only the strength of that compound. NBS performs a univariate mass testing procedure to identify those connections that exceed a statistical test threshold and belong to a specific connected component. Finally, a corrected $p$-value is calculated for each component using the null distribution of the maximum size of the connected component, which is derived empirically using a non-parametric permutation method[4,91].

To further characterize the glioma impact on the structural networks, we made use of graph-based complex network analyses on the same matrices that were used as input for TFNBS[5]. We used measures of network efficiency to detect aspects of functional integration and segregation[5]. We assessed the global efficiency[92], a measure of network integration. Global efficiency allows to assess disconnected networks, as paths between disconnected nodes are defined to have infinite length, and correspondingly zero efficiency. Additionally, we measured the local efficiency[92], a measure of network segregation. Local efficiency reflects the extent of integration between the immediate neighbors of the given node[93,94]. Furthermore, we computed measures of assortativity, degree, centrality, hierarchy, nodal efficiency, and rich club and small world organization to analyze the vulnerability and resilience of the networks and detect possible abnormalities of network connectivity[5]. These graph theoretical network analyses were performed by GRETNA 2.0.0[95].

To estimate a rank-based measure of association with topological network measures, as well as strength of edges as identified by NBS, and MRC, NIHSS, RMT ratio, tumor volume, and WHO grade, we used the FDR-adjusted Spearman rank coefficient within RStudio 1.3.1093 with R version 3.6.3. The plots were generated with the ggplot2 package[96].

Furthermore, we performed Pearson correlations to study the relationship between network measures obtained using SD_STREAM- and iFOD2-based tractography. Additional, supplementary material can be found in the Supplementary Information File.

**Ethical standard**. The study proposal is in accordance with ethical standards of the Declaration of Helsinki and was approved by the Ethics Commission of the Charité University Hospital (#EA1/016/19). All patients provided written informed consent for medical evaluations and treatments within the scope of the study.

**Reporting summary**. Further information on research design is available in the Nature Research Reporting Summary linked to this article.

# Data availability
The data that support the findings of this study are not publicly available and only available on reasonable request due to information that could compromise the privacy of the research participants.

# Code availability
The code used in this manuscript is available at: https://github.com/CUB-IGL/Network-analyses-reveal-global-and-local-glioma-related-decreases-in-ipsilesional-structural-connect. Software used: RStudio 1.3.1093 with R version 3.6.3[97].

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

## Acknowledgements

L.F., D.M., and T.P. acknowledge the support of the Cluster of Excellence Matters of Activity. Image Space Material funded by the Deutsche Forschungsgemeinschaft (DFG, German Research Foundation) under Germany´s Excellence Strategy—EXC 2025—390648296. T.R. received support from the Finnish Cultural Foundation. Figure 2 was visualized with the BrainNet Viewer toolbox (86). Funding was provided by Deutsche Forschungsgemeinschaft (DFG, German Research Foundation) under Germany´s Excellence Strategy—EXC 2025—390648296.

## Author contributions

Conceptualization: L.S.F., A.C., T.R., and B.A.; methodology: L.S.F., Z.W., D.M., S.V., T.R., and B.A.; investigation: L.S.F., Z.W., D.M., S.V., T.R., and B.A.; visualization: L.S.F., Z.W., D.M., S.V., T.R., and B.A.; funding acquisition: T.P. and P.V.; project administration: L.S.F. and T.P.; supervision: T.P. and P.V.; writing - original draft: L.S.F., Z.W., D.M., T.R., B.A., and A.C.; writing - review & editing: L.S.F., T.R., B.A., A.C., T.P., and P.V.

## Funding

## Competing interests

The authors declare no competing interests.
