## [Transparent Peer Review File · Communications Biology]

Reviewers' comments:

Reviewer #1 (Remarks to the Author):

This work investigates structural connectivity and network topology in glioma patients. The question, whether these tumors have an only local or more global impact on the white matter networks of the brain, is timely. The authors are also commended for collecting a reasonably large dataset in these patients with a rare disease, and the methodology used for diffusion MRI analysis is state-of-the-art. Furthermore, the authors make an effort to support open science by sharing (some of) their code online, though I could not find the data necessary to use the scripts available (please clarify the statement on data availability). However, the manuscript comes across as rather immature on multiple aspects, as indicated below.

General

- A major shortcoming of this network study is that no controls were included. This would not be so much of a problem if the authors did not intend to investigate the impact of glioma on the network. Technically, that is not what was investigated now: the NBS analysis merely indicates whether there are differences between hemispheres or subgroups. This approach is poorly explained throughout the manuscript, but more importantly does not fully support the conclusions drawn.
- Another pitfall of the work is the comparison between ipsi and contralateral edges. It has been clear for some time that gliomas do indeed come with remarkable alterations in structural connectivity throughout the entire brain. As such, the contralateral hemisphere does not pose a 'control' condition for the ipsilateral hemisphere. Without clear hypotheses about what differences between ipsi and contra would mean, I find it difficult to interpret the current findings.

Abstract

- The last sentence of the abstract seems like a stub and could be better integrated with the rest of the text.

Introduction

- Line 40: suggest to replace "neuronal networks" with "brain networks" to avoid confusion with deep learning approaches.
- The introduction is very lean and it may not be clear to a more general audience what a brain network is from the very brief information given on the background of this work.
- It is also not clear why the authors initially speak about brain tumors in general, but then specifically investigate glioma (grade II-IV). As far as I am aware, most if not all studies on network effects in brain tumor patients actually concern glioma patients only. It might be more straightforward to simply start with glioma.
- Line 58: "many studies" are mentioned, yet only one reference is used.

Results

- Please start the results with a brief description of the patient population studied. This information is now in the methods section, and should also be elaborated upon. What were the molecular diagnoses of these tumors (see revised WHO classification of glioma since 2016)? Did patients have epilepsy? What was their Karnofsky performance status?
- The first sentence of the results contains many abbreviations that have not been introduced. Since the Methods section is in the supplements, I suggest thoroughly rewriting the results and incorporating at least some basic statements on methodology to better guide the reader.
- Figures are generally beautiful.
- In Figure 2, it is unclear whether that these are differences between ipsi and contralesional hemispheres (if I understand correctly, but it was difficult to assess). Also, what does the color code of the nodes mean?
- Line 102 and further: please indicate whether p-values for the correlational analyses were corrected for the number of comparisons tested.

- WHO degree should be WHO grade in every instance, the authors now inconsistently use grade (correct) and degree (incorrect). Also, more conceptually: the simple grading system for glioma has been abandoned since the new classification in 2016, with particularly grade II and III gliomas being considered part of the same spectrum, for which molecular markers (IDH and 1p/19q) are much more important. This should either be taken into account or considered a limitation of the study.
- I would also advise using a different statistical approach to associate tumor subgroups (either based on grade or molecular subtype) to NOS, since subgroup is not a continuous and maybe not even an ordinal variable.
- Line 129: "Graph theoretical analysis showed significant correlations with RMT ratio, small worldness and local efficiency in contralesional hemispheres ($r_s(37) = -.373, p = .0228$)". It is unclear to me which correlates to which, particularly since three variables are mentioned (RMT ratio, small worldness and efficiency) and only one correlation statistic.
- There are multiple identical subheadings, which is very confusing.

Methods

- It is unclear whether these patients had already undergone surgery for their tumors before imaging was performed.
- Please note that the font switches in this section.
- Methods also do not specify whether any corrections were performed for the large number of correlations tested between the brain and clinical measures, and how many tests were actually performed. If no correction was performed at all, these correlations should be considered highly explorative.
- The authors should probably replace "gender" (which refers to an individuals' self-reported identity) with sex (the biological variable).

Supplementary materials

- The figures in the suppl materials are of such low resolution and small size that I cannot read them.

Reviewer #2 (Remarks to the Author):

Specific comments:

I suggest having an abbreviation guide that is easily accessible.

Title: well said, no issues.

Abstract:

Introduction: I would mention that usually graph theoretical models and connectivity analysis are based on resting state (or task-based) fMRI data. One of the innovations of the present paper is that the authors apply these methods to tractography, rather than fMRI. Otherwise, no problems.

Materials and Methods: In the future, the authors might want to work with multi-band DTI acquisition – which are far superior.

Results: The authors have a lot of clinical data, which it seems to me, they do not optimally use. For example, they have hand strength data and MEG identified localization of the hand motor homunculus. They know the extension of each tumor from the MR scans. Identifying the cortico-spinal tract should be doable in each patient. Yet, no attempt is made to incorporate the data.

Similarly, NIHSS is a very general rubric that incorporates neurological data from various systems

(strength, sensation etc). They do not get a correlation from the combined score, but it is not outside the realm of possibility that they might get correlation if they compared specific systems to specific white matter tracts (for example, the motor system).

I am having a difficult time making sense of the results in the "Probabilistic tractography" and "Deterministic tractography" sections. First it is hard to understand what is correlated to what. The "Probabilistic tractography" paragraph is all one sentence. There correlations for certain anatomical areas that almost seem random: what does the fusiform gyrus have to do with tumors centered in the pre-central (motor) gyrus? This needs to be better addressed in the Discussion also.

WHO grade, not WHO degree.

Also, would be nice to have a rigorous method to compare the two tractography methods. As thing stand, the comparison is only descriptive. Also, would be interesting to better describe (quantitate) the various difference and to see if this correlates in any way to the clinical symptoms.

Discussion: An important issue only hinted at by the authors is that DTI not only affect the connectivity but also includes other factors that decrease its reliability. For example, if the patient gets steroid that will decrease the edema (extra-cellular water) which in turn will affect all the DTI measurements. This will lead to different measurements of connectivity even though in reality, the connectivity will remain the same. This is important in differentiating the effect of gliomas tumor grade on various measurements of connectivity. Gliomas of different grades have different infiltration of the extra-axonal space (edema, glioma cells, necrosis, etc) all of which affect the DTI (and consequently connectivity) measurements. Hence, a decrease in "connectivity" described in high-grade gliomas may actually be due to increased infiltration of the extra-axonal space by edem and tumor cells, rather than due to an actual decrease in "connectivity". An imperfect way around this is to compare only tumors of one grade – but then the authors probably would not have enough patients in each group.

Figures and tables: No real issues.

Conclusion: Fine.

References: No problems.

Response to manuscript COMMSBIO-21-2462-T “Network analysis shows decreased ipsilesional structural connectivity in glioma patients”

We thank the reviewers for their valuable comments and recommendations, as well as for the opportunity to improve the quality of our manuscript. We have provided our point-to-point response to all the comments raised by the reviewers and made the required edits to the manuscript. We fully addressed their concerns and have extensively edited the manuscript based on the comments of the reviewers.

In this response document, the reviewers’ comments are displayed in **blue font**, with our responses in **bold text**. Changes made to the manuscript are noted in **italic bold text highlighted in yellow**.

Yours sincerely,
Dr. L. S. Fekonja on behalf of all authors

Reviewers' comments:

Reviewer #1 (Remarks to the Author):

This work investigates structural connectivity and network topology in glioma patients. The question, whether these tumors have an only local or more global impact on the white matter networks of the brain, is timely. The authors are also commended for collecting a reasonably large dataset in these patients with a rare disease, and the methodology used for diffusion MRI analysis is state-of-the-art. Furthermore, the authors make an effort to support open science by sharing (some of) their code online, though I could not find the data necessary to use the scripts available (please clarify the statement on data availability). However, the manuscript comes across as rather immature on multiple aspects, as indicated below.

We would like to thank the reviewer for their helpful review and for making us aware of that point. We modified our data availability statement accordingly (line no. 815): “The scripts used in this manuscript are available at: <https://github.com/CUB-IGL/Network-analyses-reveal-global-and-local-glioma-related-decreases-in-ipsilesional-structural-connect>. The data that support the findings of this study are not publicly available due to information that could compromise the privacy of the research participants.”

General

- A major shortcoming of this network study is that no controls were included. This would not be so much of a problem if the authors did not intend to investigate the impact of glioma on the network. Technically, that is not what was investigated now: the NBS analysis merely indicates whether there are differences between hemispheres or subgroups. This approach is poorly explained throughout the manuscript, but more importantly does not fully support the conclusions drawn.

We thank the reviewer for raising this point. We already address the question regarding missing controls in the Limitations section, as we also write below in relation to the

reviewer's third remark. We compare whether we can observe hemispheric differences that were caused by the tumor. Thus, we compare the ipsi- with the contralesional hemispheres. Therefore, we investigate the impact of gliomas on the hemispheric network level. We then discuss hemispheric network differences and acknowledge the limitations of comparing the ipsi- with the contralesional hemispheres. We would like to refer to Figure 2 in the manuscript, which indeed shows significant differences between the ipsi- and contralesional hemispheres, or in other words, ipsi- and contralesional networks with respect to the whole patient cohort and also to the defined subgroups. These TFNBS-based results indicate that there are indeed error-corrected significant differences between hemispheres or subgroups, even tested on the basis of two different connectome matrices: One based on a deterministic (SD_STREAM) and one based on a probabilistic (iFOD2) tractography algorithm. This is described in more detail in the results section under the section Network based statistics: *"We obtained 30 iFOD2-related and 19 SD_STREAM-related significant edges by TFNBS on the entire cohort (Fig. 2). Furthermore, we obtained 15 iFOD2-related and 14 SD_STREAM-related significant edges in the precentral subgroup and 15 iFOD2-related and 6 SD_STREAM-related significant edges in the insular subgroup (Fig. 2). Frontal and postcentral groups revealed no significant differences by TFNBS."*

Additionally, in the Results section under Network based statistics we now refer to the Materials & Method section to additionally clarify the method (line no. 87): *"To analyse structural differences between the contra- and ipsilesional hemispheres, we apply the threshold-free network-based statistics algorithm (TFNBS, see details to the method in Statistical analysis section below)."* Further, we explain the TFNBS method in more detail in the section Statistical analysis (line no. 508): *"This algorithm replaces each connectome element $M_{i,j}$ by its corresponding TFNBS score that is calculated as follows. For each connectome node i we count the number of connections $e(h)$ to other nodes j that exceed a variable threshold h . The TFNBS score is the integral of the product $e(h)^{0.4}h^3dh$ over thresholds from zero to the value of the considered element $M_{i,j}$. We performed a permutation test with a default $n=5'000$ shuffling of data for nonparametric statistical inference."*

Moreover, as we state in the reviewer's third remark below, we would like to highlight that we also discuss this topic in the conclusion section. In the Conclusion section, we correctly state that we showed altered ipsilesional connectivity in patients with unilateral gliomas. In addition, we specifically address the method and results of our study and confirm our hypothesis. In addition, we refer to further results of the study and summarize the overall observations. Furthermore, we conclude by briefly discussing the contribution of network neuroscience in relation to glioma patients and present the utility of this method regarding the present topic.

- Another pitfall of the work is the comparison between ipsi and contralateral edges. It has been clear for some time that gliomas do indeed come with remarkable alterations in structural connectivity throughout the entire brain. As such, the contralateral hemisphere does not pose a 'control' condition for the ipsilateral hemisphere. Without clear hypotheses about what differences between ipsi and contra would mean, I find it difficult to interpret the current findings.

We agree that gliomas may yield alterations not only in the ipsilesional hemisphere but in the entire brain. Nevertheless, here, we focused on measuring structural differences

between the hemispheres that are caused by the gliomas to observe hemispheric network differences. We would like to emphasize that we account for hemispherical structural differences by adding the hemispheric glioma position as a covariate as described in the Methods/Statistical analysis section: *"We added the hemispheric tumor position as a covariate to account for hemispheric differences⁴⁹".* In order to further clarify this step, we have added the following (line no. 513): ***"The hemispheres of human brains are naturally characterized by structural differences even if no lesion is present. To account for such hemispheric differences, we added the hemispheric tumor position as a covariate to account for hemispheric differences".***

In addition, in the Limitations section we discuss hemispheric network differences and indeed acknowledge the limitations of comparing the ipsi- with the contralesional hemispheres. However, despite the hemispheric differences, we clearly see tumor related significant differences (as indicated as well in Figure 2): *"Moreover, structural brain asymmetries appear to be present even in healthy subjects. A study regarding structural network topology showed that the right hemisphere network is less efficient than that of the left hemisphere⁴⁹. Specifically, the right hemisphere showed higher values of betweenness centrality and small-worldness, indicating a less optimal organization for information processing and a more random configuration compared with left-hemispheric networks. In contrast, a further study found higher global efficiency in the right hemisphere compared with the left hemispheric network⁵⁰. Finally, the small sample sizes for insular (n=8) and frontal (n=3) lobe subgroups make these data susceptible to both outliers and false negatives."*

We then further discuss this topic in the Conclusion section: *"In the present study, we showed altered ipsilesional connectivity in patients with unilateral gliomas. TFNBS analysis identified significant disconnected subnetworks involving fronto-frontal and fronto-insular connections in the ipsilesional hemisphere compared to the contralateral one. Our subgroup analysis also showed that the lesion location (e.g., pre-central and insular) affected connectivity patterns which belong to distinct and peculiar subnetworks, thus highlighting the pivotal role of lesion location in driving corresponding connectivity changes. Such connectivity changes were accompanied by reduced global and local efficiency of the ipsilesional network, suggesting tumor-related altered information transfer, which is due to the pathological involvement of both long- and short-range connectivity patterns and disturbed network integration."*

Moreover, we would like to mention that we already pose a clear hypothesis in the introduction. We start by mentioning the overall idea to introduce the reader to the research question: *"Here, we combine tractography with graph theoretical analysis and NBS to assess tumor-related structural connectome alterations within the ipsilesional hemisphere of glioma patients."*

This statement is then followed by the clearly formulated hypothesis: *"We hypothesize that asymmetries between ipsi- and contralesional connectivity profiles are related to specific tumor locations that correlate with functional impairment or neurological patient status."*

Abstract

- The last sentence of the abstract seems like a stub and could be better integrated with the rest of the text.

We agree and would like to thank the reviewer for making us aware of this shortcoming. We better integrated the last sentence of the abstract (line no. 31): ***“Network analysis showed reduced global and local efficiency in the ipsilesional hemisphere compared to the contralesional hemispheric networks, which reflect the impairment of information transfer across different regions of a network.”***

Introduction

- Line 40: suggest to replace "neuronal networks" with "brain networks" to avoid confusion with deep learning approaches.

We agree and would like to thank the reviewer for making us aware of this shortcoming. We replaced “neuronal network” with **“brain networks”** (line no. 40).

- The introduction is very lean and it may not be clear to a more general audience what a brain network is from the very brief information given on the background of this work.

We agree. We added a sentence to introduce the concept of networks in the introduction (line no. 37): ***“While classical theory assumes that local lesions have an exclusively local impact, there is increasing evidence that brain tumors yield not only local structural changes and thus may globally affect the brain¹. Modern MRI techniques such as diffusion MRI allow us to measure the structural connectivity of anatomically pre-defined brain areas, enabling the representation of brains by brain networks that contain specific brain areas as nodes and the quantified connectivity between nodes as edges.”***

- It is also not clear why the authors initially speak about brain tumors in general, but then specifically investigate glioma (grade II-IV). As far as I am aware, most if not all studies on network effects in brain tumor patients actually concern glioma patients only. It might be more straightforward to simply start with glioma.

We would like to thank the reviewer for raising this point. We now refer specifically to glioma patients in the introduction. However, we would like to point out that in fact not all studies on the effects of brain networks involve only glioma patients, but also integrate other brain tumor entities, e.g. meningiomas or pituitary tumors ¹⁻³.

- Line 58: "many studies" are mentioned, yet only one reference is used.

We would like to thank the reviewer for raising this point. We added more references (line no. 62): ***“...many other network analysis studies ¹¹⁻¹⁴,...”***

Results

- Please start the results with a brief description of the patient population studied. This information is now in the methods section, and should also be elaborated upon. What were the molecular diagnoses of these tumors (see revised WHO classification of glioma since 2016)? Did patients have epilepsy? What was their Karnofsky performance status?

We added a brief description of the studied patient population to results (line no. 74): ***“We investigated the structural network differences in 37 glioma patients (cf. Table 1; Please refer to Materials & Methods for a detailed description).”*** and refer to the now better integrated Materials & Methods section to avoid duplications.

We now describe the use of the older WHO classification as a possible limitation in the Limitations section, as requested by the reviewer and described further down in the response letter (line no. 332): ***“We would like to mention that in our study we used the***

older 2007⁵¹ WHO classification (instead of the one of 2016⁵², which considers grade 2 and 3 gliomas as part of the same spectrum for which molecular markers (IDH and 1p/19q) are much more important.”

The patients did not suffer from frequent seizures. We now state the exclusion criteria in Materials & Methods, under Patient Selection (line no. 366): **“Patients with recurrent gliomas, previous radiochemotherapy, non-glial tumors or frequent generalized seizures (more than one per week) were not considered.”**

We would like to point out that we did not make use of the Karnofsky score in this study. However, we performed the clinical assessment by using the National Institutes of Health Stroke Scale (NIHSS) to objectively quantify the impairment caused by the glioma. Besides the NIHSS, we used the British Medical Research Council grade (MRC) to assess motor status, where 0 means no muscle activation and 5 means normal muscle strength. This is stated in the Materials & Methods section, under Clinical assessment (line no. 381): **“We used the National Institutes of Health Stroke Scale (NIHSS) to objectively quantify the impairment caused by the glioma⁵³. The NIHSS includes the areas of level of consciousness, eye movements, integrity of visual fields, facial movements, arm and leg muscle strength, sensation, coordination, language, speech and neglect. Each impairment is scored on an ordinal scale of 0 to 2, 0 to 3, or 0 to 4, with the scores adding up to a total score of 0 to 42⁵³. Besides NIHSS, we used the British Medical Research Council grade (MRC) to assess motor status, where 0 means no muscle activation and 5 means normal muscle strength⁵⁴.”**

- The first sentence of the results contains many abbreviations that have not been introduced. Since the Methods section is in the supplements, I suggest thoroughly rewriting the results and incorporating at least some basic statements on methodology to better guide the reader.

We agree. We moved the Materials & Methods from the supplementary materials to the main manuscript, added a very brief statement that we analyzed the structural network in glioma patients as described in our response above and referred the reader to the now better integrated Materials & Methods section, which is no longer in the Supplementary Materials but in the main manuscript. In addition, we now provide a list of abbreviations at the end of the manuscript.

- Figures are generally beautiful.

We would like to thank the reviewer for acknowledging our figure aesthetics.

- In Figure 2, it is unclear whether that these are differences between ipsi and contralesional hemispheres (if I understand correctly, but it was difficult to assess). Also, what does the color code of the nodes mean?

We agree. We added further information to the figure legend. We have also added in the text that the node color code reflects the Desikan–Killiany–Tourville (DKT) color scheme, which enables the visual distinction of the DKT parcellation scheme (nodes; line no. 97): **“The node colors reflect the atlas parcellation (Desikan–Killiany–Tourville) color scheme.”**

- Line 102 and further: please indicate whether p-values for the correlational analyses were corrected for the number of comparisons tested.

We indeed test for multiple comparisons, as mentioned in Materials & Methods, under Statistical analysis: *“To estimate a rank-based measure of association with neuropsychological assessments, we used the FDR-adjusted Spearman rank coefficient.”*

In addition, we refer now to the Methods/Statistical analysis section (line no. 104): *“We performed Spearman correlation analyses between the strength of the edges revealed as significant by TFNBS and MRC, NIHSS, RMT ratio, tumor volume and WHO grade variables for ipsilesional, contralesional and differences between ipsi- and contralesional matrices (see Methods, Statistical analysis section below).”*

- WHO degree should be WHO grade in every instance, the authors now inconsistently use grade (correct) and degree (incorrect). Also, more conceptually: the simple grading system for glioma has been abandoned since the new classification in 2016, with particularly grade II and III gliomas being considered part of the same spectrum, for which molecular markers (IDH and 1p/19q) are much more important. This should either be taken into account or considered a limitation of the study.

We agree. We corrected degree to grade. Furthermore, we added the use of the older WHO classification to the Limitations section, as requested by the reviewer (line no. 332): *“We would like to mention that in our study we used the older 2007⁵¹ WHO classification (instead of the one of 2016⁵², which considers grade 2 and 3 gliomas as part of the same spectrum for which molecular markers (IDH and 1p/19q) are much more important.”*

- I would also advise using a different statistical approach to associate tumor subgroups (either based on grade or molecular subtype) to NOS, since subgroup is not a continuous and maybe not even an ordinal variable.

We understand the reviewer's point and agree that different statistical approaches could also be used. However, we used a tumor grade classification which is per definition an ordinal variable⁴ (the categories in the categorical variable do have natural order, but distances between categories cannot be quantified) and the Spearman correlation is the proper choice to determine the correlation between ordinal and interval variables⁵.

- Line 129: "Graph theoretical analysis showed significant correlations with RMT ratio, small worldness and local efficiency in contralesional hemispheres ($rs(37) = -.373, p = .0228$)". It is unclear to me which correlates to which, particularly since three variables are mentioned (RMT ratio, small worldness and efficiency) and only one correlation statistic.

- There are multiple identical subheadings, which is very confusing.

We agree. We clarified the provided results (line no. 136): *“Graph theoretical analysis showed significant correlations of the RMT ratio with small worldness and local efficiency in contralesional hemispheres (both $rs(37) = -.373, p = .0228$), but not with any clinical measure (i.e., MRC, NIHSS, WHO grade, tumor volume) for deterministic and probabilistic tractography results.”*

Moreover, we were not able to find multiple identical subheadings here.

Nevertheless, we have refined the two subsequent subheadings below the addressed line to (line no. 149) *“Complex network analysis in relation to probabilistic tractography”* and (line no. 156) *“Complex network analysis in relation to deterministic tractography”* to further specify the content addressed.

Methods

- It is unclear whether these patients had already undergone surgery for their tumors before imaging was performed.

We agree. We now state that that the study focuses on preoperative patient data in Materials & Methods, Patient selection and MRI data acquisition (line no. 361): *"We included n=37 left- and right-handed adult **presurgical** patients in this study (15 females, 22 males, the average age was 48.24, SD = 16.47, age range 20-78)."*

- Please note that the font switches in this section.

We would like to thank the reviewer for pointing this out. We unified the fonts.

- Methods also do not specify whether any corrections were performed for the large number of correlations tested between the brain and clinical measures, and how many tests were actually performed. If no correction was performed at all, these correlations should be considered highly explorative.

As written in the manuscript, we specify our methods to correct the correlation tests (Materials & Methods, Statistical analysis): *"To estimate a rank-based measure of association with neuropsychological assessments, we used the FDR-adjusted Spearman rank coefficient within RStudio 1.3.1093 with R version 3.6.3. The plots were generated with the ggplot2 package⁹¹."* We understand that the Materials and Methods section was part of the supplementary material and have moved it to the main manuscript for clarification, as indicated in our response above.

In addition, we would like to mention that we do indeed provide standardized information about what we tested and report the results in terms of numbers in a standardized format: $rs(df) = [r\text{-value}], p = [p\text{-value}]$, where the number after rs in parentheses corresponds to the degrees of freedom (df) and is directly related to the sample size.

- The authors should probably replace "gender" (which refers to an individuals' self-reported identity) with sex (the biological variable).

We would like to thank the reviewer for raising this timely and important topic. However, we have indeed used the patients' self-reported gender and did not test for the biological variable.

Supplementary materials

- The figures in the suppl materials are of such low resolution and small size that I cannot read them.

We agree. Unfortunately, this is related to the publisher's upload system that reduced the figures' resolutions. We can't upload larger resolution images.

However, we changed the supplementary figures, integrated Figures 5-6 now into the main text and discarded the compartmentalized plot figures due to the lack of readability, low information value and limitations regarding possible conclusions.

Reviewer #2 (Remarks to the Author):

Specific comments:

I suggest having an abbreviation guide that is easily accessible.

We agree. We now provide an abbreviation guide.

Title: well said, no issues.

We would like to thank the reviewer for the positive feedback.

Abstract:

Introduction: I would mention that usually graph theoretical models and connectivity analysis are based on resting state (or task-based) fMRI data. One of the innovations of the present paper is that the authors apply these methods to tractography, rather than fMRI. Otherwise, no problems.

We have added a sentence to the introduction and now specifically mention that we are analyzing connectomes derived from diffusion MRI data rather than functional MRI data (line no. 57):

"While many studies use functional MRI measurements to determine functional connectivity of the brain, here we use diffusion MRI (dMRI) data and analyze the resulting structural connectivity."

Materials and Methods: In the future, the authors might want to work with multi-band DTI acquisition – which are far superior.

We thank the reviewer for the hint and noted this.

Results: The authors have a lot of clinical data, which it seems to me, they do not optimally use. For example, they have hand strength data and MEG identified localization of the hand motor homunculus. They know the extension of each tumor from the MR scans. Identifying the cortico-spinal tract should be doable in each patient. Yet, no attempt is made to incorporate the data.

We thank the reviewer for acknowledging the extent of the collected data. We would like to clarify that we have not used MEG, we used TMS. We assume that the reviewer meant EMG. In an earlier work, we already studied the cortico-spinal-tract using TMS-based functional seeding ⁶. In this study however, we are interested in a connectomal approach, to find out whether there are network differences visible between the ipsi- and contralesional hemispheres (i.e. edges between nodes) and not in a distinct fiber bundle such as a TMS-motor-mapping informed CST, which is methodologically very different to the connectome approach.

However, we do make use of the TMS-derived resting motor threshold (RMT), measured using EMG, as a marker for possible motor function deterioration, which we used for our correlation tests as described in the manuscript.

In addition, we already added the Brainstem node, as specified in the Materials & Methods section (which is now in the main part of the manuscript and not anymore in the supplementary materials) to obtain as well edges between brainstem and cerebrum, reflecting edges similar to the CST: *"Furthermore, we modified the lookup table (LUT) by adding brainstem and left and right cerebellum labels from Freesurfer's LUT to FastSurfer's Desikan–Killiany–Tourville (DKT) atlas LUT to obtain nodes (left and right cerebellum and brainstem) in the connection matrix that represent the cerebral base of the primary motor area connectivity."*

Similarly, NIHSS is a very general rubric that incorporates neurological data from various systems (strength, sensation etc). They do not get a correlation from the combined score, but it is not outside the realm of possibility that they might get correlation if they compared specific systems to specific white matter tracts (for example, the motor system).

We hypothesize that asymmetries between ipsi- and contralesional connectivity profiles are related to specific tumor locations that correlate with functional impairment or neurological patient status at the connectome level (edges between nodes) and not at specific fiber bundles.

We consider the correlation from this combined score to specific white matter fiber bundles an interesting topic but out of scope: One would first need to design a new study, define which fiber bundles belong to the motor system and e.g. whether only primary motor areas should be considered. However, this would not result in a connectome-based study and strongly differ method-wise.

I am having a difficult time making sense of the results in the “Probabilistic tractography” and “Deterministic tractography” sections. First it is hard to understand what is correlated to what. The “Probabilistic tractography” paragraph is all one sentence. There are correlations for certain anatomical areas that almost seem random: what does the fusiform gyrus have to do with tumors centered in the pre-central (motor) gyrus? This needs to be better addressed in the Discussion also.

We agree and apologise for the unclear and long sentences. We split the information written in the probabilistic tractography paragraph in multiple sentences and clarified what was correlated to what (line no. 112): “Based on iFOD2 connectome matrices, there was a significant positive correlation between the streamline strength of posterior cingulate - pars opercularis and NIHSS in the contralesional hemispheres in the precentral group, $rs(13) = .79$, $p = .0238$. Furthermore, we found a positive correlation of tumor volume and inferior temporal gyrus - fusiform gyrus, $rs(8) = .90$, $p = .0301$ regarding the differences of ipsi- and contralesional matrices in the insular group. Moreover, we observed a negative correlation of tumor volume and inferior temporal gyrus – fusiform gyrus, $rs(8) = -.71$, $p = .0039$ in the ipsilesional insular group.”

Moreover, we also refined the text into multiple sentences and further clarified what was correlated to what in the deterministic tractography paragraph (line no. 121): “Based on SD_STREAM connectome matrices, there was a significant positive correlation between the weight of inferior parietal gyrus - fusiform gyrus and WHO grade regarding the difference between contra- and ipsilesional hemispheres in the insular group, $rs(8) = .88$, $p = .0226$. In addition, we observed a significant positive correlation between the streamline strength of inferior parietal gyrus and fusiform gyrus in relation to WHO grade in the contralesional hemispheres in the insular group, $rs(8) = .88$, $p = .0226$. Furthermore, we observed a significant negative correlation between the streamline strength of putamen-postcentral gyrus and WHO grade in the ipsilesional hemispheres and the entire cohort, $rs(37) = -.48$, $p = .0461$.”

However, we disagree with the reviewer’s claim regarding the discussion of the fusiform gyrus. We would like to point out that our approach does not investigate the effects of tumor to different cortical regions of the brain. As such, we have not stated that the tumor in the motor region affects the fusiform gyrus. But our findings show that certain white-matter connections projecting to the fusiform gyrus are affected. For

example, the connection between the fusiform gyrus and the inferior parietal gyrus were found to be affected by the tumor, as indicated in the text and, for example, in Figure 3.

In addition, we would like to note that we do indeed discuss these findings in detail in the discussion, pointing out the distributed organization of function but being careful not to become too speculative (line no. 223): *“Additionally, we found a significant positive correlation between streamline strength of the posterior cingulate - pars opercularis edge of the contralesional hemisphere and NIHSS in the precentral subgroup, as well as a significant positive correlation between streamline strength of inferior parietal gyrus - fusiform gyrus and WHO grade in the contralesional hemisphere in the insular subgroup. These findings may suggest that the presence of increased structural connectivity in the contralesional hemisphere is correlated both with worse clinical conditions and higher-grade gliomas, suggesting possible compensatory adaptive changes of connectomic profiles in contralesional networks in patients with unilateral gliomas^{10,35}. In addition, we found that tumor volume was correlated with the inferior temporal gyrus-fusiform gyrus edge. A larger tumor volume in the insular region likely impairs the connection strength in the ipsilesional hemisphere in this patient subgroup. Furthermore, the fusiform gyrus, particularly its anterior portion, has been shown to be strongly connected to the inferior temporal gyrus, which, as part of the human ventral visual cortex, plays a role in higher-order visual processing such as face perception and object recognition^{36,37}. Overall, our results not only demonstrate impaired ipsilesional structural connectivity in glioma patients, but also suggest that a possible neural basis of higher-order symptoms may occur due to the distributed organization of function.”*

WHO grade, not WHO degree.

We would like to thank the reviewer for making us aware of this shortcoming. We changed “WHO degree” to “**WHO grade**” throughout the manuscript.

Also, would be nice to have a rigorous method to compare the two tractography methods. As thing stand, the comparison is only descriptive. Also, would be interesting to better describe (quantitate) the various difference and to see if this correlates in any way to the clinical symptoms.

We agree with the reviewer that a quantitative analysis regarding tractography algorithm methods is interesting. We would like to point out that we indeed compare the tractography algorithms results quantitatively as demonstrated in Figure 4.

Furthermore, we performed Pearson correlations to study the relationship between network measures obtained using SD_STREAM- and iFOD2-based tractography, as stated in Materials & Methods, Statistical analysis: *“Furthermore, we performed Pearson correlations to study the relationship between network measures obtained using SD_STREAM- and iFOD2-based tractography.”*

In addition, the results illustrated in Figure 2 and Figure 3 take into account a clear quantitative description of the two tractography algorithms.

Discussion: An important issue only hinted at by the authors is that DTI not only affect the connectivity but also includes other factors that decrease its reliability. For example, if the patient gets steroid that will decrease the edema (extra-cellular water) which in turn will affect all the DTI measurements. This will lead to different measurements of connectivity even though in reality, the connectivity will remain the same. This is important in

differentiating the effect of gliomas tumor grade on various measurements of connectivity. Gliomas of different grades have different infiltration of the extra-axonal space (edema, glioma cells, necrosis, etc) all of which affect the DTI (and consequently connectivity) measurements. Hence, a decrease in “connectivity” described in high-grade gliomas may actually be due to increased infiltration of the extra-axonal space by edema and tumor cells, rather than due to an actual decrease in “connectivity”. An imperfect way around this is to compare only tumors of one grade – but then the authors probably would not have enough patients in each group.

We would like to thank the reviewer for raising this very important point.

Firstly, we would like to kindly clarify that we do not perform any diffusion tensor imaging methods. Instead, we use tractography algorithms that are based on fiber orientation distributions (FODs) that provide multiple directions per voxels as highlighted in the methods description. We agree that the steroids could cause a confounding effect. However, all patients receive preoperative steroids. Furthermore, there is evidence demonstrating that edema has no strong influence on tractography results as shown in the referenced articles ⁷.

In addition, we have added the following to the manuscript (line 321): “*Note that all patients received preoperative steroids to reduce edema, which could possibly cause a confounding effect. However, there is evidence that edema has no strong influence on tractography results as shown in the referenced articles ⁵¹”.*

Figures and tables: No real issues.

We would like to thank the reviewer for the positive feedback.

Conclusion: Fine.

We would like to thank the reviewer for the positive feedback.

References: No problems.

We would like to thank the reviewer for the positive feedback.

References used in this response letter:

- 1 Aerts, H. *et al.* Modeling brain dynamics after tumor resection using The Virtual Brain. *Neuroimage* **213**, 116738, doi:10.1016/j.neuroimage.2020.116738 (2020).
- 2 Aerts, H. *et al.* Modeling Brain Dynamics in Brain Tumor Patients Using the Virtual Brain. *eNeuro* **5**, doi:10.1523/ENEURO.0083-18.2018 (2018).
- 3 Na, S. *et al.* White matter network topology relates to cognitive flexibility and cumulative neurological risk in adult survivors of pediatric brain tumors. *Neuroimage Clin* **20**, 485-497, doi:10.1016/j.nicl.2018.08.015 (2018).
- 4 Hu, Z. D., Zhou, Z. R. & Qian, S. How to analyze tumor stage data in clinical research. *J Thorac Dis* **7**, 566-575, doi:10.3978/j.issn.2072-1439.2015.04.09 (2015).
- 5 Kathleen F. Weaver, V. M., Sarah L. Dunn, Kanya Godde, Pablo F. Weaver. in *An Introduction to Statistical Analysis in Research* 435-471 (2017).
- 6 Fekonja, L. S. *et al.* Detecting Corticospinal Tract Impairment in Tumor Patients With Fiber Density and Tensor-Based Metrics. *Frontiers in Oncology* **10**, doi:10.3389/fonc.2020.622358 (2021).

- 7 Rosenstock, T. *et al.* Specific DTI seeding and diffusivity-analysis improve the quality and prognostic value of TMS-based deterministic DTI of the pyramidal tract. *Neuroimage Clin* **16**, 276-285, doi:10.1016/j.nicl.2017.08.010 (2017).

Reviewers' comments:

Reviewer #2 (Remarks to the Author):

The manuscript reads much more nuanced now and is more mature in terms of neuro-oncological terminology and the methodology. I appreciate the extensive response to my suggestions.

I do still think that using WHO grade as an ordinal variable is slightly inaccurate. The authors apparently do not have molecular subtyping available and mention it as a limitation, which is great. However, the correlations performed may yield false results: the grades are basically an obsolete system, grade II and III are now considered a single group, where only molecular markers distinguishes between malignancies. Of course, inaccurate malignancy would only occur in very few patients, so the effect is hopefully limited. still, I would like to see a bit more background on this potential confounder to be mentioned in discussion, as it only superficially handles the issue right now. The readership of this general journal may not be familiar enough with neuro-oncology to understand exactly how this is relevant without additional explanation.

Reviewer #3 (Remarks to the Author):

The authors have answered all of my concerns. Thank you.

Response to manuscript COMMSBIO-21-2462A “Network analysis shows decreased ipsilesional structural connectivity in glioma patients”

We thank the reviewers for their valuable comments, recommendation and acceptance, as well as for the opportunity to further improve the quality of our manuscript. We have provided our response to the comment raised by reviewer #2 and made the required edit to the manuscript. We fully addressed the concern and have refined the manuscript based on this comment.

In this response document, the reviewer’s comment is displayed in **blue font**, with our response in **bold text**. Changes made to the manuscript are noted in ***italic bold text highlighted in yellow***.

Yours sincerely,
Dr. L. S. Fekonja on behalf of all authors

Reviewer's comment:

Reviewer #2 (Remarks to the Author):

The manuscript reads much more nuanced now and is more mature in terms of neuro-oncological terminology and the methodology. I appreciate the extensive response to my suggestions.

I do still think that using WHO grade as an ordinal variable is slightly inaccurate. The authors apparently do not have molecular suptying available and mention it as a limitation, which is great. However, the correlations performed may yield false results: the grades are basically an obsolete system, grade II and III are now considered a single group, where only molecular markers distinguishes between malignancies. Of course, inaccurate malignancy would only occur in very few patients, so the effect is hopefully limited. still, I would like to see a bit more background on this potential confounder to be mentioned in discussion, as it only superficially handles the issue right now. The readership of this general journal may not be familiar enough with neuro-oncology to understand exactly how this is relevant without additional explanation.

We would like to thank the reviewer for their appreciation regarding our revision and for the constructive comment.

We agree and have further revised our manuscript. We now state the additional limitations in the discussion section, as the reviewer requested: “We would like to mention that in our study we used the older 2007 54 WHO classification instead of the one from 2016 or 2021^{55,56}, which considers grade 2 and 3 gliomas as part of the same spectrum for which molecular markers (e.g. IDH, 1p/19q, TERT promoter and EGFR amplification) are of particular relevance. Thus, the correlations performed here could result in erroneous findings and should be interpreted with caution because, according to the current WHO CNS5 classification⁵⁶, neoplasms are graded within types instead across different tumor types, meaning that for instance an astrocytoma, IDH-mutant can be

grade 2, 3 or 4 . The grading of CNS tumors has long differed from other non-cerebral neoplasms, as different gradings are used for brain and spinal cord tumors ⁵⁷. WHO CNS5 has approximated the grading of non-cerebral neoplasms in the grading of CNS tumors, but has retained aspects of the traditional grading of CNS tumors, as this grading is firmly established in neuro-oncology practice. Two specific aspects of CNS tumor grading have been changed for the WHO CNS5: Arabic numerals are used (instead of Roman numerals) and neoplasms are classified within types instead of between different tumor types ⁵⁸. However, inaccurate malignancy classification would occur in very few patients in this study, and therefore this effect may be limited in the correlation analysis.”

Further changes made to the manuscript:

We specified the correlation methods in the methods section: “... as well as strength of edges as identified by NBS, and MRC, NIHSS, RMT ratio, tumor volume and WHO grade,...”

In addition, we updated the affiliation of one co-author: “⁷A.I. Virtanen Institute for Molecular Sciences, University of Eastern Finland, Kuopio, Finland.”